# Periodontitis Disease in Farmed Ruminants—Current State of Research

**DOI:** 10.3390/ijms24119763

**Published:** 2023-06-05

**Authors:** Arkadiusz Grzeczka, Marianna Lech, Gracjan Wozniak, Szymon Graczyk, Pawel Kordowitzki, Małgorzata Olejnik, Marek Gehrke, Jędrzej Maria Jaśkowski

**Affiliations:** Institute of Veterinary Medicine, Faculty of Biological and Veterinary Sciences, Nicolaus Copernicus University, 87-100 Torun, Poland; grzeczka@umk.pl (A.G.); marjanna.99@gmail.com (M.L.); wozniak@wp.pl (G.W.); graczyk@umk.pl (S.G.); p.kordowitzki@umk.pl (P.K.); mgehrke@umk.pl (M.G.); jmjaskowski@umk.pl (J.M.J.)

**Keywords:** periodontitis, cara inhada, broken mouth, cattle, goat, ovine, oral microbiome

## Abstract

Periodontal disease in ruminants is common and occurs in farmed and wild animals. Periodontal lesions can result from the secretion of endotoxins by pathogenic bacteria and as consequences of immune system activity. Three main types of periodontitis have been described. The first is chronic inflammation involving mainly premolars and molars—periodontitis (PD). The second type is an acute inflammatory reaction occurring with calcification of the periosteum of the jawbone and swelling of the surrounding soft tissues (Cara inchada, CI—“swollen face”). Finally, a third type, similar to the first but located in the incisor area, is called “broken mouth” (BM). Etiological variation between the different types of periodontitis is indicated. This particularly manifests in the composition of the microbiome, which is characteristic of the different forms of periodontitis. The widespread detection of lesions has drawn attention to the current nature of the problem.

## 1. Introduction

Animals of the suborder Ruminantia spend most of their lives grinding food with rows of flat and wide premolars and molars [1,2]. However, relatively few studies have addressed oral lesions and anatomical abnormalities in livestock. Oral lesions in horses and companion animals have been most commonly described [3,4,5,6,7,8]. However, in recent years, more attention has been paid to the oral health of livestock [9,10,11,12]. This problem particularly affects the gums, which surround the teeth, and are particularly vulnerable to mechanical damage and are often targeted by pathogenic microorganisms. However, in addition to periodontopathic bacteria, an equally important part of the pathogenesis is the inflammatory response within the oral cavity. Development of inflammation in well-vascularized and innervated gingival tissue can cause pain, tooth loss, or altered occlusion resulting in reduced food intake, decreased production, and impaired welfare [9].

There are several clinical forms of periodontitis. Depending on the species and location, periodontal disease can vary in different individuals [13]. A relevant example is a comparison of the clinical signs of cara inchada (CI) disease described in the 20th century in Brazil and periodontitis reported in ruminants in different parts of the world [14,15,16]. CI is a form periodontitis and is distinguished from the typical phenotype because it causes chronic ossifying periostitis. In addition, staff from the Embrapa Agrobiolica Institute and veterinarians from the Faculty of Veterinary Medicine in the city of Aracatuba, Brazil, demonstrated the epizootic nature of CI. CI has become a problem in ruminant farming due to intensive reclamation work of naturally forested areas. This is because areas of previously cleared tropical forests are a source of pathogenic bacteria [17]. Bacteria released from deeper layers of the soil cause acute inflammation and lead to degenerative changes that are often fatal [18]. In contrast, periodontitis (PD) in cows from European countries causes a gingival recession, alveolar atrophy, and periodontal pocket formation at the level of molars [19]. 

PD is common in breeding (cattle, sheep, goat) and companion animals. There are no reports of mortality, but the disease causes large economic losses due to reduced milk yield and an earlier decision to send the animal to slaughter [20]. In sheep and goat, periodontitis can manifest itself in a different clinical form and is called Broken Mouth (BM) [21,22]. It is also an infectious disease, but mainly causes changes at the level of the incisors and more frequent loss of the incisors [1]. Although the symptoms are concentrated in the incisors, it is also possible for inflammation to extend to the gums at the level of premolar teeth [22,23]. The various forms of periodontitis (PD, CI, and BM) have different etiologies due to the different compositions of the oral microbiome of each species and the conditions in which they live. Anaerobic bacteria of the genera *Porphyromonas* and *Prevotella* are most commonly isolated, but this is not the rule and the data presented below indicate the specificity of each condition [24,25,26,27].

Examination of the teeth and gingiva, which is an overlooked part of the clinical examination of ruminants, could provide greater control of recurring oral diseases. Detailed data have been presented on the changes in the oral microbiome of individual ruminant species. There have also been studies clarifying the course of periodontitis. The purpose of this paper is to summarize our knowledge on the incidence, etiology, pathogenesis, and consequences of the various forms of periodontitis.

## 2. Pathophysiology and Pathomorphology of the Different Forms of Periodontitis

PD is an inflammation within the periodontium as well as surrounding tissues, depending on the severity of the disease process. As a result of the accumulation of numerous bacterial strains and the immune system response, inflammation of the gums develops, later leading to alveolar recession, pocket formation, exposure of tooth necks, increased tooth mobility, and eventually tooth loss. The development of the disease is accompanied by redness, warmth, pain, and bleeding. The disease in its clinical form has been described in many animal species—domestic cattle, sheep, goats, horses, and dogs [6,9,28,29,30,31]. In addition, in laboratory studies, periodontitis was induced in rodents (mice, rats, hamsters), rhesus monkeys, dogs, cats, and Shiba goats [26,32,33,34]. 

In periodontitis, lesions are present in both hard and soft tissue. The cells forming the gingival epithelium are swollen and vacuolated [35,36]. In addition to changes in epithelial cells, inflammation is accompanied by a local accumulation of submucosal lymphoplasmatic inflammation and the presence of macrophages [36]. The contents of the periodontal pockets include food debris, a high number of neutrophils, and bacteria [36]. This causes loss of tooth attachment, increased tooth mobility, and gingival atrophy resulting in the formation of gingival pockets, and loss of collagen and bone [26,35,36]. Changes in bone tissue consist of marked remodeling and increased numbers of osteoclasts [35]. 

The cellular response to infection was found to vary depending on the bacterial species, as well as the intensity of the infection. The activity of bone tissue cells and oral soft tissue cells is manifested through increased gene expression. *P. gingivalis* has been shown to alter the expression of more genes in soft tissue cells than in bone tissue cells [37]. In contrast, *Tannerella forsythia* showed significantly higher activity in bone. Furthermore, infection with multiple bacteria resulted in a more differentiated gene response in soft tissues compared to a single infection. In contrast, no such differences were shown in bone tissue [37]. 

Polymicrobial infections can cause an increased immune system response. This is evidenced by studies that analyzed the synthesis of inflammatory mediators. The authors attributed an important role to them in the development of the disease [38,39,40,41]. Although the immune system response is beneficial to the body, the increased secretion of inflammatory mediators leads to weakened jawbones (Figure 1) [19,42,43,44]. A local increase in prostaglandins leads to increased alveolar bone loss and inhibition of collagen synthesis [27,36]. In contrast, increased concentrations of multifunctional proteins in saliva are associated with the threat response and significantly differentiate the saliva proteome of animals with periodontitis from those without periodontitis [45]. In particular, the expression of proteins related to cytoskeletal structure, humoral antimicrobial response, detection of chemical stimuli, and blood coagulation were detected.

Others believe that the destructive effect on surrounding tissues may be due to stress-related perturbations in the endoplasmic reticulum of cells exposed to bacteria. It is therefore possible that tissue destruction may be exacerbated by activation of the transduction pathway associated with the unfolded protein response (UPR) [47]. As a result of the worsening destructive processes of bone recession and developing periosteal inflammation, inflammation can amplify from a local process to a systemic response [48]. This can lead to reduced daily growth and milk production and, with inadequate nutrition, to increased inflammation [49].

Anatomopathological changes include gingival recession and periodontal pocket formation [29]. These lesions are most commonly diagnosed between the third premolar and first molar on the mandibular and maxillary bones [29]. At the level of premolars and molars, two to nine periodontal pockets are usually present along the length of the entire dental arch [29]. One of the main criteria for the diagnosis of periodontitis is the depth of the periodontal pocket, which should not exceed 5 mm in healthy individuals. Blood or purulent discharge may flow from the pockets in diseased animals [50]. Although their pathogenesis does not differ from PD, the lesions in BM are mainly concentrated in the incisal area and tooth loss occurs more frequently [1]. 

The early clinical signs of CI are similar to those of PD [51]. They first appear within the first and second premolar teeth in the mandible. As a result of bacterial enzymes such as collagenase, deoxyribonuclease, chondroitin sulphatase, fibrinolysin, gelatinase, and hyaluronidase, the architecture of the structures that provide the scaffolding for the soft tissues is disrupted [52]. As the disease progresses, it involves more teeth towards the temporomandibular joint; however, the stamen may also be affected in the incisors [53]. The disease is accompanied by periodontal epithelial ulceration and neutrophil infiltration. The purulent inflammation leads to the deepening of the periodontal pockets. When the disease starts to spread into bone tissue, there is marked atrophy of the alveolar processes due to increased osteoclastic activity and proliferation of fibrous tissue, with a consequent enlargement of the contour of the mandible. In contrast to PD, in CI, calcium accumulates within the periosteum of the mandibular bone. Ossifying periostitis develops. This manifests as a unilateral or bilateral bulging of the mandibular bone. As a result of putrefactive processes, a peculiar odor is emitted from the mouth of ruminants, which again was obtained in the laboratory when *B. melaninogenicus* was cultured in vitro [9,54].

## 3. Disease Vectors—Changes in the Microbiome in PD, BM and CI

Periodontitis is a multifactorial disease caused by a disruption of the interrelationship among the microorganisms inhabiting the oral cavity. The lesions are not the cause of the disease; they are just factors that contribute to the multifactorial nature of the disease. The physiological composition of the microbiome of the oral cavity in companion and farm animals has been determined [55,56,57]. The oral cavity of cattle showed 395 types of bacteria, and in sheep 428 types [24,25]. In comparison, 356 types of microorganisms were detected in the oral cavity of horses [55]. Substances with bactericidal properties have been detected in the saliva of ruminants so that bacterial populations remain under constant control [58]. However, in certain situations, there can be an overgrowth of pathogens. The oral microflora becomes dysbiotic and significantly differentiates itself from the healthy oral microbiome [59].

The change in bacterial populations under certain conditions contributes to the development of PD [25]. In PD, there is an expansion of pathogenic bacterial flora and a reduction in the populations characteristic of healthy individuals [25]. In inflamed oral cavity of cattle, the populations of *Elusimicrobia*, *Synergistes*, *Propionivibrio*, and *Fusobacteria* increased the most. In turn, *Gastranaerophilales*, *Planifilum*, *Burkholderia*, and *Arcobacter* are the most common populations in healthy oral cavity. 

*Elusimicrobia* is the most frequently isolated bacterium in cows suffering from PD. *Fusobacterium* has also previously been isolated from stomatitis in cows [25]. *F. nucleatum* has been identified in both the pathogenic and natural oral microbiomes [25]. Black pigmented bacteria from the genera *Prevotella* and *Porphyronomas* have been attributed with a particularly large role in the development of PD. *Prevotella* is a common cause of gingivitis in children [60]. Among the *Porphyronomas* and *Prevotella* isolated in samples from cattle with periodontitis, *Porphyronomas endodontalis* (80.7%), *Prevotella melaninogenica* (73.1%), and *Prevotella intermedia* (61.5%) predominated. In healthy cattle, *Porphyronomas endodontalis* (40%) and *Prevotella loeschei* (40%) predominated [61]. 

The genus *Treponema* is included in the stable composition of the oral bacteria of domestic cattle. In samples taken from the periapical pockets of healthy individuals, *T. amylovorum* was present in 11/14 of the samples, *T. denticola* in 11/14, *T. maltophilum* in 1/14, *T. medium* in 3/14, and *T. pectinovorum* in 9/14 of the samples [62]. The occurrence of *Treponema* in goats has also been detected during the oral disease process [26,63]. In contrast, the bacteria associated with PD in sheep belong to the genera *Petrimonas*, *Acinetobacter*, *Porphyromonas*, and *Aerococcus* [23]. Pathogens of the genus *Firmicutes* multiplied the fastest. In goats, the most frequently recorded bacteria in PD were *Fusobacterium nucleatum* (81.8%), *Tannerella forsythia* (86.3%), *Fusobacterium necrophorum* (63.6%), *Campylobacter rectus* (59.0%), *Eikenella corrodens* (45.4%), *Prevotella buccae* (31.8%), *Porphyromonas gingivalis* (18.1%), *Treponema denticola* (16.6%), *Porphyromonas endodontalis* (13.6%), and *Treponema maltophilum* (13.6%) [50]. *Porphyromonas gingivalis*, *Prevotella buccae*, and *Enterobacterales bacteria* are not observed in ruminants. However, they are a common cause of periodontitis in dogs [64,65]. 

Due to the physiological conditions of the rumen digestive system, the so-called double chewing of food, the rumen microbiota often communicates with the environment and bacteria residing in the oral cavity. However, this exchange goes both ways. Therefore, not only is the oral microbiome subject to change, but also the rumen microbiota. The rumen in healthy animals is dominated by *Bacteroidetes* (*Christensenellaceae* and *Prevotellaceae*) and *Firmicutes* (*Erysipelotrichaceae*). In diseased animals, *Bacteroidetes* (*Prevotella* and the *Ricknellaceae*) and *Firmicutes* (*Erysipelotrichaceae*) predominate [66]. 

Animals in assimilated tropical forest areas have to deal with specific pathogens that have escaped from deeper layers of the soil and are one factor in the development of oral diseases. In the case of CI, bacteria from the genera *Bacteroides*, *Actinomyces*, and *Fusobacterium* were isolated. The most frequent isolates were *Bacteroides melaninogenicus* and *Arcanobacterium pyogenes*. In the second order, *Bacteroides bivius*, *Fusobacterium nucleatum*, and *Actinomyces israelii* were identified [9,54]. A change in composition of the microbiome has been shown to influence the development of BM. A change in the oral microbiome in sheep that developed BM has been demonstrated [67]. The occurrence of potentially new phylotypes is significantly increased in samples from individuals with BM compared to samples from healthy sheep [67]. In pockets occurring in the oral cavity of sheep, *Mannheimia ruminalis* and *Moraxella caprae* were the most frequently isolated [67]. The former was particularly frequently isolated; therefore, this bacterium is considered the etiological agent of BM. However, bacteria specific to PD and CI have also been isolated in BM: *Bacteroides*, but also *Fusobacterium*, *Capnocytophaga*, *Actinomyces*, and *Veillonella* or *P. gingivalis* [25]. However, this difference may be due to less accurate methods of bacterial identification. The pathogens specific to each form are shown in table form (Table 1).

## 4. Risk Factors of PD, BM, and CI

The prevalence of PD varies between species. The prevalence of PD in goats is 50.2%, and according to others it is 70.7%. Among sheep in Scotland, 60.0% of animals had mobility or absence of one or more teeth and 87.0% of animals had periodontal pockets. In cows and calves, the PD prevalence is 12% and 27.4%, respectively [10,28,61,66]. The average age at which PD was recorded was 85.2 months in beef cattle and 102.9 months in dairy cows [29]. According to Viora et al. (2018), beef breeds are more predisposed to PD. Some beef cattle breeds suffer from PD 1.36 times more often than dairy breeds [69]. Recent reports also show the risk of disease incidence by age [10]. The disease is the most common in animals older than 6 years. In this animal category, PD was recorded in 62.5% to 100% of cows [10]. In young animals (1–4 years), the prevalence of PD ranged from 18.25% to 37.5%. 

Irrespective of breed, calculus accumulation occurs on the tooth surface of cows with increasing age. Kolenbrander et al. (2010) divided the calculus into subgingival calculus, i.e., occurring within the gingival sulcus and periodontal pockets, and supragingival calculus accumulating on the crowns of the teeth [70]. In particular, increased calculus accumulation on the incisors is a feature of BM development. The characteristic color of calculus, which is particularly visible on the crown of the tooth, is related to the metabolic activity of the bacteria inhabiting the oral cavity of ruminants.

The black-brown color of the calculus is a result of the deposition of metal elements by the biofilm [71]. Many bacteria, for example, *Porphyronomas* and *Prevotella*, use iron in their metabolism. Animals with particularly intense calculus accumulation are more likely to have PD and BM or be in the development stage of the disease [72]. The rate of calculus formation is also highly dependent on the type of diet. Increased amounts of silicic acid salts in the diet promote easier calculus deposition [73]. Recently, it has been shown that there are differences in the incidence of periodontitis episodes between groups fed hay or hay with the addition of a protein supplement [45]. 

The anaerobic bacteria that lead to tartar formation in the genus *Prevotella* and *Porphyronomas* multiply much more intensively in a site with limited oxygen. For this reason, the development sites for PD and BM can be small injuries caused, for example, by eating feed that is too hard [74]. The animals, in addition to feeding on the given feed, are grazed. This factor appeared to be the determining factor in the occurrence of CI. In the 1970s and 1980s in Brazil, it was popular to graze cattle and smaller ruminants on assimilated tropical forest land [15,75]. Clearing these areas released anaerobic bacteria from deeper in the soil, leading to the development of purulent stomatitis in some animals. The effects were so persistent that feeding fodder made from grasses growing on previously rainforest land led to the development of CI [53,70,71]. As a result, the etiological agent of the disease has been attributed to millet. In addition, calves that had been fed milk from cows grazing in areas covered with great millet also contracted CI [75]. Agricultural practices in these areas, which involved the use of small amounts of antibiotics, also contributed to the disruption of the oral microbiome, promoting the growth of invasive bacteria [54]. In 2004, the etiologies of CI were clarified and summarized in the three subsections cited below:

“(1) Cattle affected are at the age when premolar and molar teeth erupt; (2) bacteria of the Bacteroides group are present in the subgingival spaces; (3) the ingestion with the forage of subinhibitory concentrations of antibiotics, mainly streptomycin, produced by the large increase in the number of actinomycetes found in soils from pastures sown after recently cleared forest; this leads to an increased adherence of *Bacteroides* spp. to the gingival epithelium and to the progressive destruction of the periodontal tissues.” [76].

## 5. Consequences of PD, BM, and CI

PD, BM, and CI lead to reduced feed intake and thus reduced productivity [77]. In daily practice, PD is rarely diagnosed, which may be the reason for the animals’ reduced feed intake. A lower caloric supply in the daily diet has a direct effect on the reduced production capacity of the mammary glands [78]. In addition, difficulties with chewing lead to weight loss and exacerbate the problem. For breeders of beef cows, this is of particular importance because the fattening period until slaughter is precisely defined. In turn, in herds of dairy goats affected by periodontitis, there is a period of reduced productivity and weight loss until it is cured [79]. In the case of calves suffering from CI, they were transported to other farms to reduce exposure to the etiological agents. The transport costs significantly reduced the profitability of breeding [15]. In the final stages of the disease, even treated animals were dying. 

The occurrence of periodontitis, even a mild version of it, can deprive a significant percentage of individuals in the herd of a comfortable life. The perception of pain by animals is difficult to define due to species-specific behavioral conditions [80]. Consequently, there is the possibility of animals experiencing undetected pain due to periodontitis [81]. In other cases, pain occurs only in the setting of an acute inflammatory reaction or in the presence of necrotic lesions [82]. However, it is difficult to directly transfer results from human studies due to the previously highlighted difficulties in recording pain in animals. 

## 6. Treatment of PD, BM, and CI

Two types of chemotherapeutics are used to control the periodontitis. Of the drugs, non-steroidal anti-inflammatory drugs (NSAIDs) are the most studied. The indication for the use of this group of drugs is their anti-inflammatory effect. NSAIDs exert an inhibitory effect on cyclooxygenase (prostaglandin-endoperoxide synthase), thereby reducing the immune system response. The reduction in the synthesis of cell mediators dulls the inflammation at the site of infection. An additional advantage of these drugs is that they can be administered topically as well as systemically [83,84]. Research on NSAIDs began in the 1970s. The first drug tested was indomethacin [85]. Its therapeutic effects have been demonstrated in dogs, rodents, and primates [85,86]. Indomethacin reduced bone recession and resulted in reduction in the incidence of this disease following its use. Similar efficacy was demonstrated by flurbiprofen used in combination with ibuprofen [84]. In addition, treatment with naproxen and piroxicam has been shown to have satisfactory effects [83,87]. The treatment regimen of non-steroidal anti-inflammatory drugs in ruminants has not been developed, so studies are needed on the use of NSAIDs for cows. 

In domestic cattle, antibiotics have been used to treat periodontitis [8,18]. One study used an antibiotic belonging to the macrolide group—spiramycin [88]. It is currently used to treat oral, urinary, and respiratory infections. The use of spiramycin in humans led to complete silencing of active periodontitis and a 30% reduction in the depth of periodontal pockets [89,90]. In the case of domestic cattle, spiramycin administered prophylactically together with mineral supplements completely prevented the occurrence of periodontitis [88]. However, such an approach is not in line with regulations in many countries, including European Union member states [91]. 

The impact of virginiamycin, a streptogramin antibiotic, was studied in 2019 [8]. Ongoing administration of virginiamycin reduced the incidence of periodontitis in the study group (31 cases) compared to the control group (58 cases). Similar results were obtained in another study, in which the frequency of periodontitis was lower in virginiamycin-treated calves compared to the control group [92]. However, the use of virginiamycin is restricted in European Union member states due to its placement in category A [93]. 

A study in dogs showed that the use of ampicillin, amoxicillin with clavulanic acid, gentamicin, imipenem, and vancomycin reduced the number of periodontal lesions in dogs [64,65]. In contrast, clear levels of resistance were observed with chloramphenicol (10%), erythromycin (20%), and streptomycin (35%). However, this species has a different oral microbiome, so it is uncertain if these results can be extrapolated to ruminants.

In addition to their antibacterial action, tetracyclines have an anti-inflammatory effect, thus inhibiting bone resorption and facilitating bone regeneration [93,94,95]. An important feature of the drugs used in oral treatment is their assimilation into the tooth surface [96]. Tetracyclines by themselves do not have such properties. However, carriers have been developed for this group of antibiotics. One of the cyclic oligosaccharides used is cyclodextrin [97]. The tetracycline and β-cyclodextrin complex formed a protective surface for the exposed part of the tooth [97,98]. The bactericidal effect of this complex was maintained for three months. However, the antimicrobial effect of tetracyclines is attenuated by eight classes of genes encoding ribosome-protective proteins: tet(M), tet(O), tet(P), tet(Q), tet(S), tet(W), and tet(T) [99,100,101]. In some cases, the development of resistance by 95% of the population has been described [64,102]. Therefore, the high level of bacterial resistance to tetracyclines has made these antimicrobials useless. 

Azithromycin, an antibiotic with anti-inflammatory effects, appears promising for the treatment of periodontitis [103]. Azithromycin has been shown to have this effect both in vitro and in vivo. The susceptibility of periodontal bacteria to a wider range of antimicrobial drugs was tested in 1985 in cattle suffering from CI [9,18]. However, the use of macrolides in farming animals should be restricted to life-threatening diseases, because they are valuable class of antibiotics for animal and human treatment [104]. Taking into account the high incidence of PD and long treatment periods, the use of macrolides, just like other antimicrobials, could promote resistance.

Some of the above results must be taken with caution because periodontitis is a multifactorial disease, and the results obtained from in vitro studies are not necessarily reflected in clinical cases. Use of antibiotics is limited also by antimicrobial stewardship issues. Even in human medicine, the use of antimicrobial in treatment of PD is not recommended [105].

As an alternative to the above approaches, surgery may involve excision of the sinus flap to remove the infected tooth [106]. After its removal, the resected bone flap was placed back in its anatomical position, and the cow was protected with preoperative NSAIDs such as meloxicam or flunixin meglumine and perioperative antibiotic prophylaxis with ampicillin sodium. The size of ruminant teeth may make it impossible to remove them intact, so it is sometimes necessary to dissect the tooth so that it can be removed safely [107]. However, more similar treatments are required to fully confirm the efficacy of the above method. Additionally, subgingival scaling and root planning, as well as a surgical approach, which are standard procedures in the treatment of periodontitis in humans and companion animals, are not available in ruminants because of low time- and cost-efficiency [105]. 

Another effective method to reduce the effects of IC was described in Brazil. Calves that suddenly contracted IC were transferred to other farms where IC was not observed, resulting in the cessation of disease symptoms. A factor that can positively influence the microbiota is the presence of a female with the calf. It is not a determinant of recovery, but may be useful in the healing process, as indicated by differences in the fecal microbiota of calves reared with or without a female [108]. 

## 7. Conclusions

Several negative effects associated with the occurrence of PD, BM, and CI can reduce animal welfare. Treatment of these changes is based on the use of anti-inflammatory drugs. However, these measures only alleviate the symptoms. The applicability of antibiotics is limited due to antimicrobial stewardship and insufficient data. Wider studies on bacterial resistance have only included epizootic forms of periodontitis, namely CI [8,18]. PD, BM, and CI are diseases with complex etiologies whose clinical manifestations can easily be overlooked during clinical examination. Unfortunately, there is still little knowledge about the management of these types of periodontal diseases in ruminants [10,25]. Time and cost efficiency is also a limiting factor when proposing lines of treatment.

## Figures and Tables

**Figure 1 ijms-24-09763-f001:**
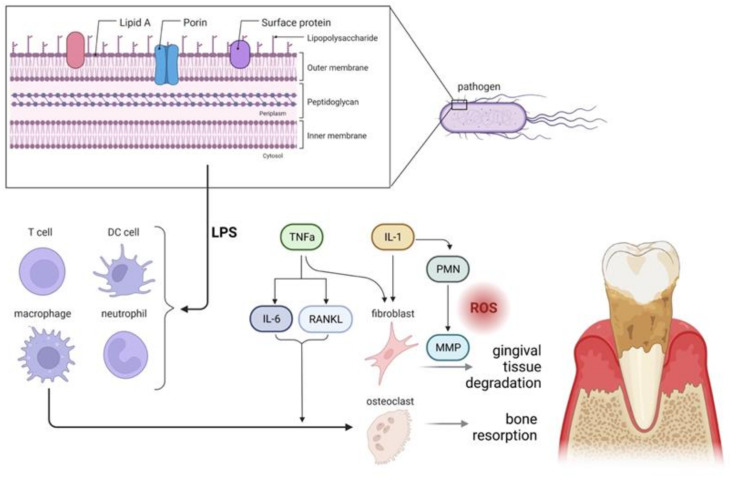
Bacterial lipopolysaccharide (LPS) and cell wall elements of pathogens trigger immune system responses. This leads to an increased release of cytokines (IL-1, TNF-a) from local tissues and immune cells. IL-1 activates immune cells and facilitates production of reactive oxygen species (ROS) through degranulation of polymorphonuclear leukocytes (PMNs). Increased production of inflammatory mediators, e.g., prostaglandin (PGE2) and matrix metalloproteinase (MMP), contributes to impaired collagen synthesis resulting in the degradation of gingival tissues. TNF-a, on the other hand, is the main signal for cell apoptosis, bone resorption, MMP secretion, and IL-6 production. IL-6 stimulates osteoclast formation (activation of the RANK/RANKL/OPG axis) and promotes bone resorption [19,42,43,44,46].

**Table 1 ijms-24-09763-t001:** Table showing bacteria differentiating oral conditions [9,23,24,25,67,68].

	Isolation of Different Bacteria Depending on the Specific Disease That Has Been Diagnosed			
	Healthy	Mild Version of Periodontitis (PD)	Cara Inchada	Broken Mouth
*Bovine*	*Gastranaerophilales, Planifilum, Burkholderia, Arcobater, Escherichia-Shigella, Pseudomonas, Actinobacteria,*	*Elusimicrobia, Synergistes, Propionivibrio, Fusobacteria, Wolinella, Porphyromonas endodontalis, Porphyromonas asaccharolytica, P. melaninogenica, P. intermedia P. asaccharolytica, Candidatus,* *Prevotella buccae, Prevotella intermedia, Prevotella melaninogenica and Prevotella oralis, Firmicutes, Bacteroides, Methanomethylophilus,* *Treponema denticola, Treponema amylovorum e Treponema maltophilum, Clostridium, Mollicutes, Desulfuromonas, Stenotrophomonas, Peptostreptococcus*	*A. pygenes, Bacteroides, Fusobacterium nucleatum and Actinomyces israell*	*Not applicable*
*Ovine*	*Pasteurellaceae, Neisseria, Fusobacterium, Pseudomonas, Porphyromonas, leptotrichiaceae, Enterobacter, Hafnia alvei*	*Tannelrella forsythia, Treponema denticolica,* *Petrimonas, Fusobacterium necrophorum, Fusobacterium nucleutum, Acinetobacter,* *Porphyromonas gingivalis, P. endodontalis,* *Aerococcus, Bacteroides, Christensenellaceae, Pervotella,* *Treponema amylovorum, T. denticola, T. maltophilum, Treponema medium e Treponema pectinovoru, Prevotella buccae, Prevotella melaninogenica, Prevotella nigrescens, * *Ent erococcus faecium*	*Petrimonas, Fusobacterium, Acinetobacter, * *192 Porphyromonas, Aerococcus, Bacteroides, Fastidiosipila, Succiniclasticum, Peptostreptococcus, tenerella,* *193 Christensenellaceae R7, Fretibacterium*	*Mannheimia ruminalis, Moraxella caprae, Clostridiales bacterium, Moraxella cuniculi, Cloacibacterium* sp., *Micrococcus luteus, Actinobcillus pleuropneumoniae, Anaerococcus* sp., *Manheimia glucosida, Treponema amylovorum, Treponema denticolica, Treponema maltphilum, Treponema medium, Treponema pectinovorum*
*Goat*	*Fusobacterium nucleatum, Tannerella forsythia, Actinomyces israelii, Pervotella nigrescens, Enterococcus faecium, Fusobacterium necrophorum, Campylobacter recrus*	*Fusobacterium nucleatum, Tannerella forsythia, Fusobacterium necrophorum, Campylobacter rectus, Eikenella corrodens, Pervotella buccae, Actinomyces israelli, Porphyromonas gingivalis, Pervotella nigrecescens, Pervotella loescheii, Treponema denticola, Porphyromonas endodontalis, Treponema maltophilum, Treponema melaninogenica*	*Not applicable*	*Not applicable*

## Data Availability

Not applicable.

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
