# Peer review of "Periodontitis Disease in Farmed Ruminants—Current State of Research"

_ijms, 2023, doi:10.3390/ijms24119763_

Round 1
Reviewer 1 Report
MAJOR ISSUES
The purpose of this article was to summarize knowledge on the incidence, etiology, pathogenesis and consequences of the three types of periodontal disease (periodontitis, broken mouth and cara inchada) in ruminants – in short, to present the current state of research on the topic.
In my opinion, this review article needs major revision before being acceptable for publishing in IJMS.
Chapter 2 (Pathophysiology and pathomorphology of the different forms of periodontitis) should be restructured. The authors should clearly present if there are any differences between lesions in PD and BM on a cellular/tissue level. Secondly, the authors should avoid presenting well-known facts about the involvement of major inflammatory mediators (paragraphs 5 & 6 of chapter 2) because these can be generally applied to inflammatory response and do not make PD or BM distinguishable entities.
Additional issue is about the references cited in chapter 6 (Treatment). Namely, two thirds of references are dated between 1979 and 1998. The of the article says „...current state of research“. To use mostly dated references in the closing (and one of the most important chapters of the article) does not present the current but past state of research. Accordingly, it also leaves the impression that nobody in this particular field of research cares even though PD, BM and CI cause significant economic losses. Therefore, my recommendation is to expand this chapter by introducing references from novel research within the last 10 years.
Also, the conclusion (lines 329-330) is rather confusing – different microbiological backgrounds of PD and BM are not the cause of little knowledge about the treatment of these conditions. Please elaborate what that might be.
MINOR ISSUES:
1. Line 16-17 (Abstract) – periodontitis is not a side effect of the immune system, but rather a consequence of the host's immune response; please rephrase the statement accordingly.
2. Line 34 (introduction) – instead of „whose are“ put „and are“.
3. Line 41 (introduction) – instead of „some clinical forms of periodontitis“ put „several clinical forms of periodontitis“.
4. Line 46 (Introduction) – typo, remove extra „In addition“.
5. Line 55-56 (Introduction) – please add reference/s to the statement about the economic losses due to reduced milk yields from cattle affected by periodontitis.
6. Line 60 (Introduction) – typo, remove extra space in front of punctuation mark.
7. Line 64 (Introduction) – is it Porphyromona or Porphyromonas?
8. Line 66 (Introduction) – replace „gingivens“ with „gingiva“.
9. Line 82 (chapter 2) – be specific abot what „this condition“ referes to. If it is about PD, put „PD“ instead.
10. Line 86-87 (chapter 2) – clarify what does „2 to 9“ refer to?
11. Line 99-100 (chapter 2) – please remove this sentence. It is redundant.
12. Line 122 (chapter 2) – do not use „tartar“ and „calculus“ interchangeably, my recommendation is to use the term „calculus“. Please standardize throughout the text.
13. Line 130 (chapter 2) – is it Porphyromona or Porphyromonas?
14. Line 142 (chapter 2) – remove „that have formed“ to make the sentence more clear.
15. Line 143 (chapter 2) – rephrase, instead of „begins to evolve“ put „starts to spread into“.
16. Line 147 (chapter 2) – remove „In addition“ at the beginning of the sentence.
17. Lines 154 & 159 (Figure 1 caption) – use abbreviations for IL-1 and IL-6 for interleukin-1 and -6.
18. Line 163-164 (chapter 3) – sentence beginning with „The lesions are not the cause...“ is unlcear anr redundant so best remove it.
19. Line 173 (chapter 3) – remove „conditions“.
20. Line 175-176 (chapter 3) – rephrase the sentence like this: „In inflamed oral cavity of cattle, the populations of ... increase the most.“
21. Line 176-177 (chapter 3) – rephrase the sentence like this: „In turn, ... are the most frequent populations in healthy oral cavity.“
22. Line 179 (chapter 3) – remove „shown in a 2018 study“.
23. Line 186 (chapter 3) – remove „(n=25)“. If the reader is more interested, he/she can check the details in ref. 60
24. Line 198-199 (chapter 3) – change the font size. Remove italic where appropriate.
25. Line 215 (chapter 4) – change the numbering of this chapter in the title. It should be „4. Risk factors of PD, BM and CI“. You should update the numbering for the rest of the chapter titles inlcuding „7. Conclusions“.
26. Line 219 (chapter 4) – typo, remove extra space from „is12%“.
27. Line 227 (chapter 4) – standardize, use „calculus“ instead of „tartar“.
28. Line 228-230 (chapter 4) – rephrase the sentence because it is unlcear.
29. Line 245 (chapter 4) – typo, replace „subinibitory“ with „subinhibitory“.
30. Line 250 (chapter 5) – grammar, replace „leads to“ with „lead to“.
31. Line 250-251 (chapter 5) – please rephrase the sentence because it is unlcear.
32. Lines 261, 264, 269, 284, 287, 289 & 315 (chapter 5 & 6) – used abbreviation „PD“ instead of „periodontitis“.
33. Line 278 (chapter 6) – please remove „a“ from „resulted in a reduction“.
34. Line 281-282 (chapter 6) – use abbreviation „NSAIDs“.
35. Line 286-287 (chapter 6) – please rephrase the statement about the effectiveness of spiramycin. How can periodontitis be completely cured if the pocket depth is reduced by 30%? Do you mean that spiramycin can be used to reduce the inflammation of periodontal tissue?
36. Line 288-289 (chapter 6) – this statement is also unlcear. Are you referring to prophylactic use of spiramycin?
37. Line 312 (chapter 6) – instead of „it“ put „Azithromycin“.
38. Line 316-317 (chapter 6) – please explain why the treatment of oral diseases in ruminants by macrolides is not recommended. This paragraph begins with the statement that the use of Azithromycin for the treatment of PD is promising.
39. Line 318-320 (chapter 6) - transfer of animals to other farms in case of CI can be effective for the treatment of the disease, but it is not cost-effective as you mentioned previously in the text (see lines 257-260).
The English language in the manuscript needs editing.
Reviewer 2 Report
Reviewer
Manuscript ID: ijms-2395864
Type of manuscript: Review
Title: Periodontitis disease of farmed ruminants - current state of research
Authors: Arkadiusz Grzeczka *, Marianna Lech, Gracjan Wozniak, Szymon
Graczyk, Pawel Kordowitzki, Małgorzata Olejnik, Marek Gehrke, Jędrzej Maria
Jaśkowski
Submitted to section: Molecular Pathology, Diagnostics, and Therapeutics,
https://www.mdpi.com/journal/ijms/sections/Pathology_Diagnostics_Therapeutics
Comments. The article updates knowledge about periodontal diseases specific to ruminants. It should be remembered, several particularities that characterize these animals and which can have an influence on the oral microbiota. On the other hand periodontal disease always follows the same scenario and needs to be better explained. Finally, therapeutic means remain limited insofar as hygienic therapy is impossible.
It should be emphasized that cattle present severe periodontitis with strong family aggregation (accumulation of cases in the same family, genetic origin) combined with shared environmental factors (specific pathogens, regular hygiene impossible).
The particularity of ruminants being regurgitation, it is obvious that the composition of the digestive microbiota will greatly influence that of the oral cavity and vice versa?! Therefore, it is necessary to detail the composition of the intestinal microbiota complexity.
Ruminal digestions are carried out by bacteria, protozoa, archaea and fungi. The rumen hosts a diverse microbiota: approximately 200 species of bacteria (1010 to 1011 bacteria per mL), protozoa (104 to 106 per mL) and fungi (between 103 and 105 zoospores per mL – zoospores are flagellate motile spores participating in the reproduction of certain Eumycetes). There are also between 107 and 109 bacteriophage virus particles per mL.
A periodontal disease whatever the individual begins with gingivitis (superficial infectious inflammation of the mucous membrane of the periodontium) then this one evolves in depth by attacking the alveolar bone we then speak of periodontitis. Bleeding, bone loss, pocket depth are the progressive signs of this disease accompanied by clinical signs (redness, heat and pain).
The dental and ruminal microbiomes of cattle with periodontitis and clinically healthy have different profiles. The results suggest the presence of a dysbiotic community and an inflammatory environment in the dental biofilm and rumen of cattle with periodontitis. Borsanelli AC, Athayde FRF, Riggio MP, Brandt BW, Rocha FI, Jesus EC, Gaetti-Jardim E Jr, Schweitzer CM and Dutra IS (2022) Dysbiosis and predicted function of dental and ruminal microbiome associated with bovine periodontitis. Front. Microbiol. 13:936021. doi: 10.3389/fmicb.2022.936021
For more precision . The initial treatment for any periodontal disease is to sanitize the oral cavity which is impossible on the scale of an entire herd. Then, in case of persistence of the disease, drug therapy is considered as a second intention. The objective is to restore or maintain the balance of the oral microbiota.
The immune reaction in relation to the microbial plaque is briefly treated.
The proposal of a specific microbial plaque for periodontal disease according to the three former proposed categories does not seem obvious.
Clinical diagnosis of periodontal disease should be made according to recent parameters used by Borsanelli 2022 and not on old descriptions.
Adding references.
. Borsanelli AC, Athayde FRF, Riggio MP, Brandt BW, Rocha FI, Jesus EC, Gaetti-Jardim E Jr, Schweitzer CM and Dutra IS (2022) Dysbiosis and predicted function of dental and ruminal microbiome associated with bovine periodontitis. Front. Microbiol. 13:936021. doi: 10.3389/fmicb.2022.936021
Zhang, Z.; Huang, B.; Wang, Y.; Zhu, M.; Wang, C. Could Weaning Remodel the Oral Microbiota Composition in Donkeys? An Exploratory Study. Animals 2022, 12, 2024. https://doi.org/10.3390/ ani12162024
Differences in the fecal microbiota of dairy calves reared with differing sources of milk and levels of maternal contact.
Beaver A, Petersen C, Weary DM, Finlay BB, von Keyserlingk MAG. JDS Commun. 2021 Jun 23;2(4):200-206. doi: 10.3168/jdsc.2020-0059. eCollection 2021 Jul. PMID: 36338447 Free PMC article.
Round 2
Reviewer 1 Report
I have reviewed the authors’ response and corrections introduced in the manuscript. There are still some minor issues that need to be addressed before the quality of manuscript is acceptable for publishing in IJMS.
1. Lines 115-116 (Chapter 2) – please remove these lines because you list here the factors related to innate immunity (TLRs) and most common inflammatory mediators (IL-1β. TNF-α, IFN-γ…). These can be seen active in any kind of inflammatory response and are not specific for PD, BM, or CI. The paragraph should be focused on specifics.
2. Lines 118-121 (Chapter 2) – here you mentioned differences in saliva proteome of healthy animals and animals with PD due to increased concentration of multifunctional proteins. Can you be more specific? Please, elaborate.
3. Lines 340-342 (Chapter 6) – please rephrase the sentence. Prophylaxis refers to a treatment given, or action taken to prevent disease, not to eliminate the disease. I understood what you wanted to say, but it still needs to be written clearly.
4. Lines 377-379 (Chapter 6) – the sentence is non sequitur and should be rephrased. So why is it that macrolides are not used for the treatment of PD in animals? Are they commonly used for the treatment of other animal diseases? And, if the PD is so prevalent and would require extensive use of macrolides, are we not afraid of microbes developing resistance to macrolides? Try to rephrase this section along these lines.
5. Line 405 (Chapter 7) – please remove the opening sentence of this paragraph which begins with “Reducing the incidence of periodontitis…”
6. Lines 413-415 (Chapter 7) – I would break this down into two sentences. Maybe like this: “PD and BM are diseases with complex etiologies whose clinical manifestations can easily be overlooked during clinical examination. Unfortunately, there is still little knowledge about the management of these types of periodontal diseases in ruminants.” May I also ask – what about CI? Why did you omit Ci from this section?
7. Line 413 (Chapter 7) – use abbreviation “CI” instead of “cara inchada”.
Minor editing of English language required.
Author Response
Arkadiusz Grzeczka Toruń, 27.05.2023
Marianna Lech
Gracjan Wozniak
Szymon Graczyk
Paweł Kordowitzki
Małgorzata Olejnik
Marek Gehrke
Jędrzej Maria Jaśkowski
Institute of Veterinary Medicine
Nicolaus Copernicus University in Toruń
Toruń, Poland
Dear Reviewer,
Thank you very much for your pertinent comments, which will help us to improve the text.
Ad. 1
Lines 115-116 (Chapter 2) – please remove these lines because you list here the factors related to innate immunity (TLRs) and most common inflammatory mediators (IL-1β. TNF-α, IFN-γ…). These can be seen active in any kind of inflammatory response and are not specific for PD, BM, or CI. The paragraph should be focused on specifics.
Respond:
We agree. We removed sentence.
Ad. 2
Lines 118-121 (Chapter 2) – here you mentioned differences in saliva proteome of healthy animals and animals with PD due to increased concentration of multifunctional proteins. Can you be more specific? Please, elaborate.
Respond:
We have listed the main functions that have been attributed to the identified proteins. We have not listed them all because there are too many of them, so it misses the point. the inquisitive reader can use the work cited.
Ad. 3
Lines 340-342 (Chapter 6) – please rephrase the sentence. Prophylaxis refers to a treatment given, or action taken to prevent disease, not to eliminate the disease. I understood what you wanted to say, but it still needs to be written clearly.
Respond: We agree. Redacted to make it clearer.
Ad. 4
Lines 377-379 (Chapter 6) – the sentence is non sequitur and should be rephrased. So why is it that macrolides are not used for the treatment of PD in animals? Are they commonly used for the treatment of other animal diseases? And, if the PD is so prevalent and would require extensive use of macrolides, are we not afraid of microbes developing resistance to macrolides? Try to rephrase this section along these lines.
Respond: We agree. Redacted to make it clearer.
Ad. 5
Line 405 (Chapter 7) – please remove the opening sentence of this paragraph which begins with “Reducing the incidence of periodontitis…”
Respond:
We agree. We removed sentence.
Ad. 6
Lines 413-415 (Chapter 7) – I would break this down into two sentences. Maybe like this: “PD and BM are diseases with complex etiologies whose clinical manifestations can easily be overlooked during clinical examination. Unfortunately, there is still little knowledge about the management of these types of periodontal diseases in ruminants.” May I also ask – what about CI? Why did you omit Ci from this section?
Respond:
We agree.
Ad. 7
Line 413 (Chapter 7) – use abbreviation “CI” instead of “cara inchada”.
Respond:
We agree.
With kind regards,
All Authors
Reviewer 2 Report
I reread the manuscript. The authors have made the requested changes.Best regards.
Author Response
Arkadiusz Grzeczka Toruń, 27.05.2023
Marianna Lech
Gracjan Wozniak
Szymon Graczyk
Paweł Kordowitzki
Małgorzata Olejnik
Marek Gehrke
Jędrzej Maria Jaśkowski
Institute of Veterinary Medicine
Nicolaus Copernicus University in Toruń
Toruń, Poland
Dear Reviewer,
Once again, we thank you very much for your pertinent comments. We believe that they have significantly improved the quality of the article.
With kind regards,
All Authors